# SARS-CoV-2 Infection as a Possible Trigger for IgA-Associated Vasculitis: A Case Report

**DOI:** 10.3390/children10020344

**Published:** 2023-02-10

**Authors:** Edyta Machura, Helena Krakowczyk, Katarzyna Bąk-Drabik, Maria Szczepańska

**Affiliations:** Department of Paediatrics, Faculty of Medical Sciences in Zabrze, Medical University of Silesia, 40-055 Katowice, Poland

**Keywords:** Henoch–Schönlein purpura, IgAV, SARS-CoV-2, children

## Abstract

Background: IgA-associated vasculitis (IgAV), formerly known as Henoch–Schönlein purpura (HSP) disease, is the most common type of systemic vasculitis observed during developmental age. Available published studies associate the outbreak of the disease with streptococci, adenovirus, parvovirus, mycoplasma, respiratory syncytial virus (RSV), and influenza infection in approximately 50% of patients with HSP, while some emerging reports have described a few cases of COVID-19 infection being associated with HSP in both adults and children. Case presentation: a 7-year-old girl was diagnosed with HSP, fulfilling the four required clinical criteria (palpable purpura and abdominal pain, arthralgia and edema, and periodic renal involvement). Infection with SARS-CoV-2 was confirmed via the presence of IgM and IgG antibodies. The disclosure of the Henoch–Schönlein purpura (HSP) disease was preceded by a mild, symptomatically treated infection of the upper respiratory tract. High levels of inflammatory markers were observed during hospitalization, including leukocytosis, an increased neutrophil count and a high neutrophil-to-lymphocyte ratio (NLR). All of these markers are associated with IgAV gastrointestinal bleeding, which was also associated with rotavirus diarrhea observed in the patient. Conclusions: This case presented by us and similar cases presented by other authors indicate the possible role of SARS-CoV-2 in the development of HSP, but this assumption requires further research and evidence-based verification.

## 1. Introduction

IgA-associated vasculitis (IgAV), formerly known as Henoch–Schönlein purpura (HSP) disease, is the most common type of systemic vasculitis observed during developmental age. According to new nomenclature established during the International Chapel Hill Consensus Conference 2012 (CHCC 2012), it is included in the group of inflammatory diseases of the small vessels associated with immune complex deposits [1]. The onset of the disease most often occurs between the ages of 3 and 15, with peak incidence at the age of 4–6 years with an estimated incidence of 20.4/100,000 children per year. The incidence rate of the disease, observed in the autumn and winter months, may suggest a relationship with infectious factors, especially with a history of upper respiratory tract infections [2]. So far, no trigger factor of the disease has been identified. A cross-response to bacterial and viral antigens likely causes the disease. Other possible triggers may include drugs (ACEI, antibiotics, NSAIDs), vaccines (against influenza, hepatitis A and B), toxins (e.g., following insect bites), food allergens, and cancer antigens (lung, breast, prostate, esophageal, or colon cancer and lymphomas) [2,3,4,5]. In IgAV, deposits of immune complexes that mainly consist of immunoglobulin A1 (IgA1) and the complement component C3 plus neutrophil infiltrates are found inside the walls of small vessels (mainly capillaries, venules, and arterioles). Recent reports have suggested that, in this disease, immune complexes str formed by IgA with the soluble receptor for this immunoglobulin (FcαRI), and abnormal glycosylation of the IgA1 hinge region occurs, causing the formation of the so-called galactose-deficient IgA1 (Gd-IgA1), against which IgG autoantibodies may develop [2]. Immune complexes typically affect the skin, gastrointestinal tract, and musculoskeletal system. Red blood cells, while leaking into the skin, lead to typical cutaneous hemorrhages, which are a hallmark of IgAV. The clinical manifestations of IgAV may vary in their severity and the order of their occurrence, which may result in either a delayed or wrong diagnosis. Recent consensus criteria for the diagnosis of HSP in childhood require the presence of palpable purpura with at least one of the following symptoms: abdominal pain, histopathology showing immunoglobulin A (IgA) deposition, arthritis/arthralgia, or renal involvement [1]. In approximately 50% of patients with HSP, an outbreak of the disease was associated with streptococci, adenovirus, parvovirus, mycoplasma, RSV, or influenza infection [3,4].

Severe acute respiratory syndrome coronavirus 2 infection (SARS-CoV-2) was identified for the first time in Wuhan, China in December 2019 and has since quickly spread across the entire globe. Overall, children younger than 18 years globally account for only 1% to 2% of detected COVID-19 cases [6]. In contrast to adults, younger children demonstrate a milder clinical course of the disease, with shorter time periods to the resolution of the symptoms and lower infectivity. The reason for this remains unknown, but it is possible that the lower number of ACE2 receptors in children’s nasal mucosae limits viral entry, while higher levels of ACE2 receptor in the lungs protect against the virus [7]. This virus may, however, induce serious complications in conjunction with both acute infection and other associated phenomena, such as multisystem inflammatory syndrome in children (MIS-C), which is fortunately rarely reported [6]. Some emerging reports of cases observed in both adults and children described a few records of COVID-19 infection associated with HSP [8,9,10]. Gastrointestinal symptoms, such as abdominal pain, and skin symptoms have frequently been reported in COVID-19 infections. Moreover, a possible association of an IgA vasculitis-like effect with SARS-CoV-2 infection has been suggested. Anti-SARS-CoV-2 IgA is the first immunoglobulin to be detected after a COVID-19 infection [11]. In addition, an exacerbated, IgA-mediated humoral response observed in COVID-19 patients may predispose them to the formation of IgA complex deposits in the vascular endothelium and to the development of IgA vasculitis [8].

Here, a case of a 7-year-old girl with immunoglobulin A vasculitis and COVID-19 is presented. She was diagnosed with HSP, fulfilling the following four clinical criteria: palpable purpura and abdominal pain, arthralgia and joint oedema, and periodic renal involvement. The patient was admitted to the general pediatric department for further medical evaluation.

## 2. Case Report

A 7-year-old previously healthy girl was admitted to the general pediatric department for the appearance of petechial skin lesions located mainly in the lower limbs, and for pain in and the swelling of multiple joints (see Figure 1A–C). Two weeks before hospitalization, the girl demonstrated symptoms of a mild infection of the upper respiratory tract. Being fever-free, she was symptomatically treated, while other family members were in good health at that time.

On admission, the girl was in good general condition and without fever. A physical examination revealed a petechial rash typical of Henoch-Schönlein purpura in the areas of the ankles, feet, and shins with swelling accompanied by heat and limited mobility in the ankle joints, the left knee joint, and in the wrists and fingers; limited mobility in the right knee and in the right elbow joints; dryness of the oral mucosa; tenderness of the abdominal wall around the navel on palpation. Laboratory tests (see Table 1) revealed increased inflammatory parameters (CRP, WBC with lymphocytic smear), elevated D-dimers, acetonuria, and a single episode of proteinuria. An antigen test for SARS-CoV-2 was negative on admission. A chest X-ray and an ultrasound of the abdominal cavity were also performed, demonstrating no significant abnormalities. The clinical scoring scale of IgAV, modified by Fessatou S. et al., was 3 on admission. This clinical scoring is the sum of points depending on the severity of joint, renal, and gastrointestinal symptoms (mild course ≤ 4 points; severe > 4 points) [12].

Treatment included prophylactic antibiotic therapy (cefuroxime intravenously), parenteral hydration, and vascular sealing medication (cyclonamine, rutoside). Due to the intensification of abdominal pain and the presence of occult blood in the stool test in the days following admission, treatment with prednisone was added at a dose of 1 mg/kg/day with good results, following abdominal pain relief. On the 10th day of hospitalization, a sudden deterioration of the general condition was observed in the evening and night hours, with violent vomiting and massive bleeding from the lower gastrointestinal tract (numerous abundant diarrheal stools mixed with fresh blood). An ultrasound examination of the abdominal cavity revealed areas of intussusception within the small intestine (in the interpretation of the ultrasound of the abdominal cavity: fragmentary visualization of intestinal loops, segmentally swollen up to 7–9 mm in the area of the small intestine, containing significant amounts of loose food contents and presenting oscillating movements). The descending colon, sigmoid colon, and rectum were filled with loose contents, revealing fresh blood and signs of gastrointestinal obstruction in the abdominal examination (in the left and middle abdomen, several short levels of fluid were visible; however, no free gas was visible under the diaphragm domes). Laboratory test results (see Table 1) revealed an increase in inflammatory parameters (CRP: 55.5 mg/l, procalcitonin: 2.38 ng/mL, WBC: 38,000/uL, with a granulocytic smear), a decrease in red-cell parameters (HGB: 9.2 g/dL, Ht: 26%), and the consumption of coagulation parameters (fibrinogen: 208 mg/dl, kaolin-cephalin time: 22.7 s, prothrombin time: 12.7 s). The presence of rotaviral infection was also confirmed with a rapid immunochromatographic test for the qualitative detection of antigens. The girl presented with a fever of up to 38.5 °C; her vital signs were: HR: 180 beats/min, O_2_ Sat 98%, BP: 90/65 mmHg. The patient was assessed by a surgeon with no indications for emergency surgical intervention. A conservative treatment was implemented: intensive parenteral hydration, and antiemetic and cytoprotective drugs. Methylprednisolone pulses of 30 mg/kg were administered for three consecutive days with a broad-spectral antibiotic (carbapenem) and a single infusion of fresh frozen plasma. During several subsequent hours, hemodynamic stabilization was achieved (HR: 100–110 beats/min, BP: 100/60 mmHg, O_2_ sat 98%) with an improvement of the general condition. Blood count parameters were stable and further monitored, while the girl passed some loose stools with some amount of fresh blood.

Due to the current epidemiological situation in Poland and the possible occurrence of a multisystem inflammatory syndrome associated with COVID-19 infection, laboratory diagnostics were extended to test antibodies for SARS-CoV-2, obtaining reactive results in both classes of antibodies. The concentration of specific neutralizing antibodies for the spike protein-S antigen (spike protein) of the SARS-CoV-2 virus was assessed. The patient was not vaccinated for COVID-19. The laboratory-tested parameters obtained during the hospitalization period are shown in Table 1. During the following days in hospital, systemic steroid therapy was continued, and no disturbing symptoms were observed. After 17 days of stay in hospital, the child was discharged home in a good general condition, with normal blood count values, negative inflammatory markers, absent proteinuria, normal BP parameters and with a recommendation to gradually reduce doses of the systemic steroids. She was advised to stay under the care of gastroenterological and nephrological control in an outpatient setting. After three weeks, a cardiovascular evaluation was performed with echocardiography to exclude post-COVID-19 complications. After that time, the laboratory test results were normal, and IgA concentrations lowered to 1.9 g/L, reaching the normal value.

## 3. Discussion

Regarding the presented case, we considered the assumed role of SARS-CoV-2 a trigger for HSP. IgA vasculitis associated with COVID-19 infection was only documented in a few children [10,14,15,16,17]. Four human coronaviruses, namely, (hCoVs)-229E, HKU1, NL63 and OC43, circulate globally, typically cause mild upper respiratory infections, and may be possible triggers of HSP in children [3]. Moreover, in some previous reports, hCoV NL63 was implicated as a likely trigger pathogen of acute hemorrhagic edema of infancy (AHEI), a benign type of small-vessel leukocytoclastic vasculitis with perivascular IgA deposition that is seen in one-third of AHEI cases [18]. In addition, a link was reported between COVID-19 and Kawasaki disease, which is another type of childhood vasculitis [19]. Regarding our case, the observed skin changes were preceded by a mild infection of the upper respiratory tract, which was treated symptomatically. The negative bacteriological test results and low ASO titer values excluded *Streptococcus pyogenes* infection with high probability, which, apart from viruses, is considered one of the main triggers of HSP. On admission, the high values of inflammation markers such as leukocytosis, increased neutrophil count, and high NLR were significant predictors of systemic involvement in HSP [13]. It was also suggested that the higher NLR and PLR values were the markers associated with IgAV gastrointestinal bleeding, which was also observed in the girl. Despite the administered treatment, a sudden deterioration of the child’s condition happened on the 10th day of hospitalization; massive gastrointestinal bleeding associated with anemia and increased inflammatory parameters were observed, associated with rotavirus diarrhea (a nosocomial infection). Rotaviruses may also be the cause of HSP [3], but due to the short period (1–3 days) of their incubation, they may not have been the case in our patient.

Prednisone, at a dose of 1 mg, was initially used for moderate abdominal pain and occult blood in the stool, and was then replaced with intravenous methylprednisolone, applied in accordance with the guidelines [20]. In two cases of pediatric IgAV after COVID-19 infection similar to ours, massive bleeding from the gastrointestinal tract was observed.

HSP is most often characterized by a mild, self-limiting course and a good prognosis, but, in some cases, serious complications may also develop. Symptoms from the gastrointestinal tract (GT) affect 50–70% of patients, most often with abdominal pains, nausea, vomiting or latent bleeding. Hematemesis, gastrointestinal bleeding, intussusception, bowel ischemia with secondary necrosis, or bowel perforation is less frequently identified [2,13]. A severe course of the disease may also be associated with neurological complications. Our patient did not present such symptoms. Infection with SARS-CoV-2 before admission to the hospital was the most likely cause of the symptoms that presented in the following days on the basis of the results, which was confirmed via the presence of both IgM and IgG antibodies in our patient, with a tendency for seroconversion in terms of IgM antibodies. In some children in the acute phase of COVID-19 disease, IgM antibodies or antibodies in both classes were found at the same time. Characteristics of the immune response in pediatric patients were examined, and confirmed the rapid production of protective antibodies and the rapid switching of classes of produced immunoglobulins from IgM to IgG, even within 1 week of exposure to SARS-CoV-2. This effective humoral response could explain the mild or absent symptoms of infection in infected children, as in our patient [21,22]. It was not possible to determine anti-SARS-CoV-2 antibodies in the IgA class in the patient. The IgA serum concentration (2.99 mg/L) was also initially increased and then decreased after 3 weeks of treatment. According to the available reports, IgA is present in the serum of COVID-19 patients. Levels of Ag-specific IgA were markedly increased approximately 2 weeks after the onset of symptoms, and remained continuously elevated for the subsequent 2 weeks. It appears from clinical observations that relative levels of IgA are markedly higher in severe patients compared with nonsevere patients.

IgA is traditionally recognized to play an anti-inflammatory role and preventing tissue damage at mucosal sites [11]. However, recent reports have also demonstrated that serum IgA is involved in the formation of immune complexes to amplify inflammatory responses by inducing proinflammatory cytokine and chemokine production through various cells, including macrophages, dendritic cells, monocytes, and Kupffer cells [23]. A severe COVID-19 infection may, at least in part, be an IgA-mediated disease (related to IgA deposition and vasculitis), which helps in explaining commonly observed organ injuries in COVID-19 infections (kidney injury, acute pulmonary embolism, etc.) [11].

## 4. Conclusions

The presented case and presentations of other authors [10,14,15,16,17] indicate a possible role of SARS-CoV-2 in the development of HSP, but this assumption requires further research and thorough verification.

## Figures and Tables

**Figure 1 children-10-00344-f001:**
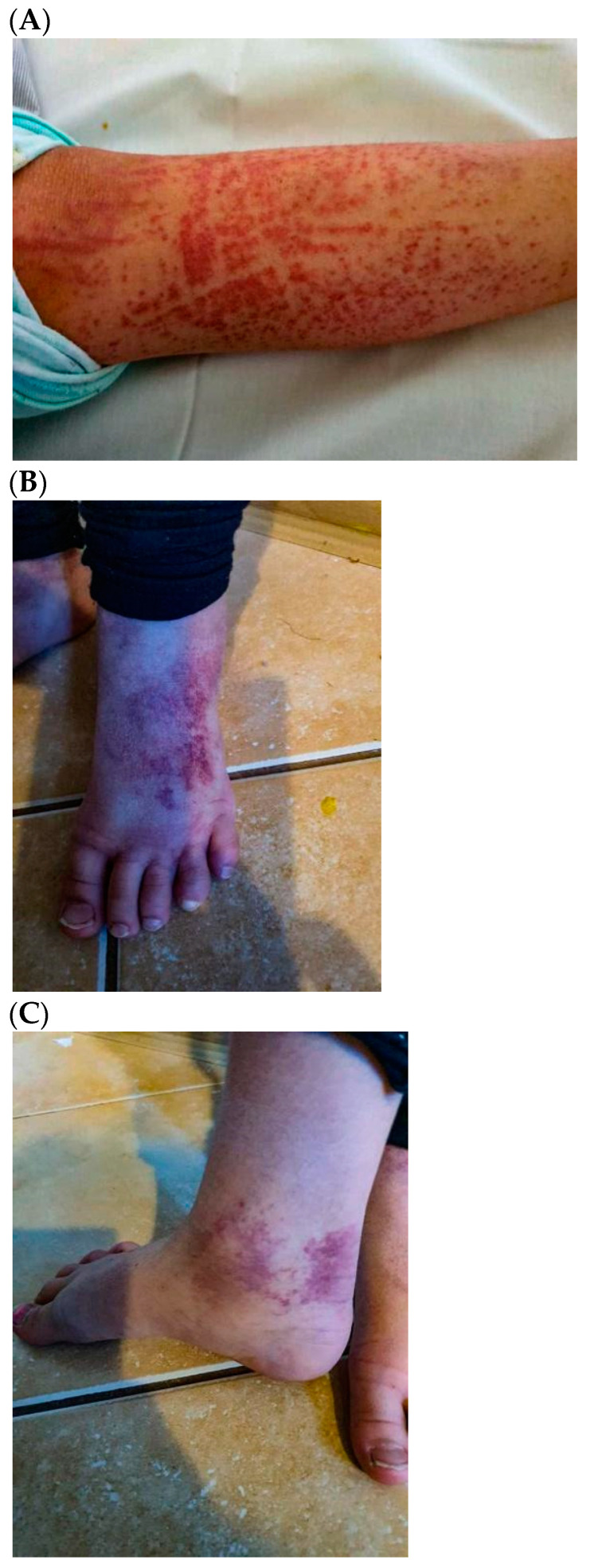
(**A**–**C**) Rash in the girl’s lower limb and ankle with characteristic HSP edema.

**Table 1 children-10-00344-t001:** Laboratory parameter values of the index patient during hospitalization.

Parameter	Result		
	Hospitalization Beginning	Deterioration	Hospitalization End
Hemoglobin (g/dl)	14.4	9.2	12.8
Total leukocyte count (cells/µL)	15.63	38	24.84
Differential count (%)/(cells/µL)			
Neutrophils	73.6/11.52	82.2/31.26	66.3/16.5
Lymphocytes	16.5/2.58	14.1/5.32	21.3/5.28
Platelets (cells/µL)	418	477	571
Neutrophil-to-lymphocyte count ratio (NLR) (1–1.91) [13]	4.46	5.86	3.12
Platelet-to-lymphocytes count ratio (PLR) (95.47–152.32) [13]	162.01	433.66	108.14
Mean platelet volume divided by platelet count (MPR) (0.029–0.037) [13]	0.022	0.018	0.016
IgA (g/L) (0.33–2.35)	2.9		
IgM (g/L) (0.36–1.98)	1.89		
IgG (g/L) (8.53–14.4)	11.45		
C3 (g/L) (0.9–1.8)	1.21		
C4 (g/L) (0.1–0.4)	0.23		
24 h urinary protein (mg)	1800		
Serum creatinine (mg/dL)	0.45	0.38	0.34
Sodium (mEq/L)	138	133	141
Potassium (mEq/L)	4.42	4.86	4.73
Aspartate aminotransferase (IU/L) (0–40)	27.1	22.2	16.1
Alanine aminotransferase (IU/L) (0–41)	8.6	10.8	21.6
Protein (g/dL) (5.7–8.2)		5.7	7.3
Albumin (g/dL) (3.8–5.4)		3.49	4.4
Prothrombin time (11–16)	15.4	12.5	15.5
Kaolin-kephalin time (28–40)	31.8	22.7	28.8
Fibrynogen mg/dL (200–400)	449	208	309
INR (0.8–1.2)	1.16	1.7	0.95
C-reactive protein (mg/L) (0–5)	21.93	55.5	0.99
d-Dimer ug/mL (0–0.5)	9.12	3.92	0.73
Procalcitonin (ng/mL) (0–0.5)		2.38	0.08
Rapid SARS-CoV-2 antigen test	Negative		
SARS-CoV-2 IgM positive > 1.00 index		3.2	
SARS-CoV-2 IgG positive > 7.1 BAU/mL		132.67	

## Data Availability

The data presented in this study are available on request from the corresponding author.

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
