# Peer review of "SARS-CoV-2 Infection as a Possible Trigger for IgA-Associated Vasculitis: A Case Report"

_children, 2023, doi:10.3390/children10020344_

Round 1

Reviewer 1 Report

Authors have preseted the case of  IgA-associated vasculitis in SARS-CoV-2 infection.

This is very important case in terms of clinical relevance.

Authors have confirmed the SARS-CoV-2 by Antibody testing and co-relate as well with the sympotomes.

The observation and assumption made based on the previous reports where similar cases were reported.

Article needs grammer and sentence synthesis changes.

Author Response

Drogi Recenzencie           

ChcielibyÅ›my podziÄ™kować za wszystkie uwagi, które byÅ‚y bardzo pomocne w ulepszeniu naszego artykuÅ‚u i przygotowaniu ostatecznej propozycji manuskryptu do publikacji.          

Artykuł został poprawiony i zmieniony zgodnie z sugestią. Zasugerowano, aby rękopis został poddany gruntownej rewizji w języku angielskim. Korzystaliśmy z płatnej usługi oferowanej przez Dział Redakcji Angielskiej MDPI. Zaświadczenie zostało dołączone do wniosku.

 W imieniu wszystkich autorów

Z pozdrowieniami,

Heleny Krakowczyk

Reviewer 2 Report

The authors present a case of a 7 y.o girl who presented with IgAV, 2 weeks after a presumed SARS-COV-2-infection. The case is alright presented, however needs editing with regards to punctuation and English language. Moreover, the prodromal phase of upper resp. tract infection is not obviously associated to a Sars-Cov_2 infection, as the antigen test was negative, and the patients already was seroconverted from IgM to IgG. Hence, if the current case report should reach the level of publication, the link between Sars-Cov-2 and the subsequent IgAV diagnosis must be more firmly established. 

Questions/issues

Abstract line 10-11, 14-15, 20, 70-71: sentence needs editing

Abstract line 16 and 72. The authors state that there was no renal involvement. That is wrong - there was renal involvement as she presented with proteinuria!

Laboratory results

What sort of Sars-Cov-2 Ab were measured -spike or nucleocapsid?

Can the authors establish a more solid link between the infection and Sars-Cov-2?

Author Response

Drogi Recenzencie,

       ChcielibyÅ›my podziÄ™kować za wszystkie cenne uwagi, które byÅ‚y bardzo pomocne w udoskonalaniu naszego artykuÅ‚u i przygotowaniu ostatecznej propozycji manuskryptu do publikacji.

       ArtykuÅ‚ zostaÅ‚ poprawiony i zmieniony zgodnie z sugestiami. DokÅ‚adnie doÅ‚ożyliÅ›my wszelkich staraÅ„, aby wprowadzić zmiany. Konkretne odpowiedzi znajdziesz poniżej.

W imieniu wszystkich autorów

Z pozdrowieniami,

Helena Krakowczyk

Odpowiedzi na pytania i uwagi Recenzentów:

        Zasugerowano, aby rÄ™kopis zostaÅ‚ poddany gruntownej rewizji w jÄ™zyku angielskim. KorzystaliÅ›my z pÅ‚atnej usÅ‚ugi oferowanej przez DziaÅ‚ Redakcji Angielskiej MDPI. ZaÅ›wiadczenie zostaÅ‚o doÅ‚Ä…czone do wniosku

  1. Autorzy przedstawiajÄ… przypadek 7-letniej dziewczynki, która zgÅ‚osiÅ‚a siÄ™ z IgAV 2 tygodnie po podejrzeniu zakażenia SARS-COV-2. Sprawa jest dobrze przedstawiona, wymaga jednak korekty w zakresie interpunkcji i jÄ™zyka angielskiego. Ponadto faza prodromalna górnych wzgl. infekcja przewodu pokarmowego nie jest oczywiÅ›cie zwiÄ…zana z infekcjÄ… Sars-Cov_2, ponieważ test antygenowy byÅ‚ ujemny, a pacjenci byli już serokonwertowani z IgM na IgG. StÄ…d, jeÅ›li obecny opis przypadku miaÅ‚by osiÄ…gnąć poziom publikacji, zwiÄ…zek miÄ™dzy Sars-Cov-2 a późniejszÄ… diagnozÄ… IgAV musi zostać mocniej ustalony. Dyskusja mogÅ‚aby być bardziej przejrzysta i zwiÄ™zÅ‚a.

Odpowiedź:   Poprawione w tekÅ›cie rÄ™kopisu. Sekcja dyskusji - wiersz 179-188, strona 4.

Zakażenie wirusem SARS-COV2 przed przyjÄ™ciem do szpitala byÅ‚o najbardziej prawdopodobnÄ… przyczynÄ… objawów wystÄ™pujÄ…cych w kolejnych dniach na podstawie uzyskanych wyników, co zostaÅ‚o potwierdzone obecnoÅ›ciÄ… przeciwciaÅ‚ IgM i IgG u naszej pacjentki, z tendencjÄ… do serokonwersji pod wzglÄ™dem przeciwciaÅ‚ IgM. U niektórych dzieci w ostrej fazie choroby COVID-19 stwierdzono jednoczeÅ›nie przeciwciaÅ‚a IgM lub przeciwciaÅ‚a w obu klasach. Zbadano charakterystykÄ™ odpowiedzi immunologicznej u pacjentów pediatrycznych i potwierdzono szybkÄ… produkcjÄ™ przeciwciaÅ‚ ochronnych oraz szybkÄ… zmianÄ™ klas wytwarzanych immunoglobulin z IgM na IgG, nawet w ciÄ…gu 1 tygodnia od ekspozycji na SARS-Cov2. Ta efektywna odpowiedź humoralna może wyjaÅ›niać Å‚agodne lub nieobecne objawy infekcji u zakażonych dzieci, jak u naszego pacjenta [20,21].

  1. Abstrakt 10-11, 14-15, 20, 70-71: zdanie wymaga przeredagowania

Odpowiedź: Poprawione w tekście rękopisu

  1. Streszczenie linii 16 i 72. Autorzy stwierdzają, że nie było zajęcia nerek. To jest złe - było zajęcie nerek, kiedy prezentowała białkomocz!

Odpowiedź:  U 7-letniej dziewczynki rozpoznano HSP, speÅ‚niajÄ…c cztery wymagane kryteria kliniczne (plamica wyczuwalna palpacyjnie i bóle brzucha, bóle i obrzÄ™ki stawów oraz okresowe zajÄ™cie nerek).

Sekcja streszczenia – wiersz 16, strona 1

Część wstÄ™pna – wiersz 74, strona 2

  1. Jakiego rodzaju Sars-Cov-2 Ab zmierzono – kolce czy nukleokapsydy?

Odpowiedź: Oceniono stężenie przeciwciaÅ‚ neutralizujÄ…cych swoistych dla biaÅ‚ka wypustki – antygenu S (biaÅ‚ka wypustki) wirusa SARS-CoV-2.

Sekcja opisu przypadku – linia 123-124, strona 3.  

  1. Czy autorzy mogą ustalić bardziej solidny związek między infekcją a Sars-Cov-2?

Odpowiedź:   Naszym zdaniem najbardziej prawdopodobnÄ… przyczynÄ… zachorowania przed przyjÄ™ciem do szpitala byÅ‚o zakażenie wirusem Sars-Cov 2. Objawy zakażenia byÅ‚y Å‚agodne i wymagaÅ‚y jedynie leczenia objawowego. Przy przyjÄ™ciu do szpitala wykonano badania bakteriologiczne górnych dróg oddechowych, które daÅ‚y wynik ujemny, nie potwierdzono podwyższonego miana ASO (część dyskusji – wiersze 158-159, str. 4). Potwierdzono natomiast podwyższone stężenia przeciwciaÅ‚ IgM i IgG swoistych dla zakażenia Sars-Cov2 u pacjenta nieszczepionego (szczegóÅ‚owe wyjaÅ›nienie w odpowiedzi na pytanie 1)

Reviewer 3 Report

it a good case presentation .

Author Response

Drogi Recenzencie,

DziÄ™kujemy za wszystkie pozytywne komentarze, które byÅ‚y bardzo pomocne w przygotowaniu ostatecznej propozycji manuskryptu do publikacji. ArtykuÅ‚ zostaÅ‚ poprawiony i zmieniony zgodnie z sugestiÄ….

W imieniu wszystkich autorów

Z pozdrowieniami,

Heleny Krakowczyk